# Purification Method of Extracellular Vesicles Derived from Human T-Cell Leukemia Virus Type 1-Infected Cells without Virions

**DOI:** 10.3390/v16020249

**Published:** 2024-02-04

**Authors:** Katsumi Kawano, Yuki Hashikura, Kunihiko Umekita

**Affiliations:** 1Division of Respirology, Rheumatology, Infectious Diseases and Neurology, Department of Internal Medicine, University of Miyazaki, Kihara 5200, Kiyotake, Miyazaki 889-1692, Japan; katsumi_kawano@med.miyazaki-u.ac.jp; 2Clinical Laboratory, University of Miyazaki of Hospital, Kihara 5200, Kiyotake, Miyazaki 889-1692, Japan; yuuki_hashikura@med.miyazaki-u.ac.jp

**Keywords:** extracellular vesicles, human T-cell leukemia virus type 1, virions, Tim-4 affinity purification, genomic RNA

## Abstract

To mediate intercellular communication, cells produce extracellular vesicles (EVs). These EVs transport many biomolecules such as proteins, nucleic acids, and lipids between cells and regulate pathophysiological actions in the recipient cell. However, EVs and virus particles produced from virus-infected cells are of similar size and specific gravity; therefore, the separation and purification of these two particles is often controversial. When analyzing the physiological functions of EVs from virus-infected cells, the presence or absence of virus particle contamination must always be verified. The human T-cell leukemia virus type 1 (HTLV-1)-infected cell line, MT-2, produces EVs and virus particles. Here, we validated a method for purifying EVs from MT-2 cell culture supernatants while avoiding HTLV-1 viral particle contamination. EV fractions were collected using a combination of immunoprecipitation with Tim-4, which binds to phosphatidylserine, and polymer precipitation. The HTLV-1 viral envelope protein, gp46, was not detected in the EV fraction. Proteomic analysis revealed that EV-constituted proteins were predominant in this EV fraction. Furthermore, the EVs were found to contain the HTLV-1 viral genome. The proposed method can purify EVs while avoiding virus particle contamination and is expected to contribute to future research on EVs derived from HTLV-1-infected cells.

## 1. Introduction

Extracellular vesicles (EVs), mediate intercellular communicationand are involved in various processes such as immune responses [1]. Based on particle size, composition, and developmental mechanism, EVs are categorized into exosomes, microvesicles, and apoptotic bodies. EVs are commonly formed of a lipid bilayer membrane with exposed phosphatidylserine. Four-transmembrane proteins called tetraspanins are abundantly expressed on the membrane surface of exosomes and microvesicles, and molecules such as CD9, CD81, and CD63 are considered general EV markers [2]. EVs also contain proteins, lipids, nucleic acids, etc. [2]. EVs, after being extracellularly secreted, are suspected to travel to adjacent or distant tissues through bodily fluids such as blood, urine, and saliva, and function as mediators of intercellular communication [3,4,5].

EVs collected from various biological samples such as the plasma, urine, and cell culture supernatant are used for investigating EV functions. EVs have also been found in cerebrospinal fluid of patients with HTLV-1-associated myelopathy [6,7]. EV-collecting methods have recently been developed, with some common methods being ultracentrifugation, density gradient centrifugation, polymer precipitation, and ultrafiltration based on the particle size and specific gravity of EVs. Immunoprecipitation is another method in which specific proteins present in EV membranes are targeted [8]. Although studies have analyzed the function of EVs produced by various cells, virus-infected cells produce both EVs and virus particles [9,10]. Therefore, based on the collection method used, the purified EV fraction may contain virus particles. The International Society for Extracellular Vesicles proposed Minimal Information for Studies of Extracellular Vesicles guidelines for the field in 2014. Although the MISEV2018 guidelines include tables and outlines of suggested protocols and steps to be followed to document specific EV-associated functional activities, no detailed recommended protocol has been presented for collecting and purifying EVs from virus-infected cells [11]. Several reports have suggested that EVs derived from virus-infected cells contain viral nucleic acids and proteins and that they play a role in viral infection [8,12,13]. However, EVs and virions commonly have a particle size of approximately 100 nm in diameter, and limitations may exist in EV separation and purification based on their particle size and density gradients. EVs derived from virus-infected cells may coexist with virus particles cannot be denied, and standard protocols to separate them must be developed for the biological functional analysis of the derived EVs.

Human T-cell leukemia virus type 1 (HTLV-1)-infected cells produce virions and EVs in the culture supernatant. These virions are approximately 100 nm in size, and the viral envelope is an infected cell-derived lipid bilayer membrane. Gp46 and gp21 are envelope proteins. Virions contain the Gag protein and two copies of the viral genome RNA. The Gag protein is composed of the matrix protein p19, the capsid protein p24, and the nucleocapsid p15 [14]. HTLV-1-infected cell-derived EVs activate T cells [15]. They also induce IFNγ production in T cells [15]. Several studies reported that EVs derived from HTLV-1-infected cells contain HTLV-1-associated proteins and their mRNAs [6,16,17]. Both the Ultracentrifugation and the Nanotrap^®^ (NT) particle method were previously used for EV isolation, but HTLV-1 virions may be present in the purified EV fraction obtained using these methods [16]. HTLV-1-infected cell lines produce HTLV-1 virions and EVs. Therefore, to functionally analyze EVs derived from HTLV-1-infected cell lines, a method is required for purifying EVs and HTLV-1 virions with higher purity.

Here, we used MT-2 cells, a HTLV-1-infected cell line that produces HTLV-1 virus particles, and EVs. This study established a method for purifying EVs derived from HTLV-1-infected cells from the MT-2 culture medium. A protocol for separating HTLV-1 virus particles and the derived EVs by combining gravity gradient centrifugation and Tim-4 immunoprecipitation was established. Moreover, the purified EVs contained HTLV-1 genomic RNAs.

## 2. Materials and Methods

### 2.1. Cell Culture

The HTLV-1-infected cell line MT-2 was purchased from the Japanese Collection of Research Bioresources Cell Bank (JCRB1210, Lot No.09142007, Ibaraki, Japan). MT-2 cultures were maintained in RPMI1640 supplemented with 10% fetal calf serum (FCS) and 1% penicillin/streptomycin (PSN). To prepare the EV-free culture medium, RPMI1640 supplemented with 10% FCS and 1% PSN was ultrafiltrated using Amicon Ultra-15 Centrifugal Filter Devices (Merck KGaA, Darmstadt, Germany). MT-2 was cultured in the EV-free culture medium in each experiment.

### 2.2. Isolation of EVs

First, 1.0 × 10^6^/mL MT-2 was cultured in a tissue culture dish (100 mm × 20 mm; FALCON) for 72 h. The MT-2-cultured medium was centrifuged for 5 min at 400× *g*, 4 °C to remove cell debris. The cell culture supernatant was collected and centrifuged for 30 min at 2000× *g*, 4 °C. To remove apoptotic bodies and large-sized microvesicles, the supernatant was filtrated using the Spritzen-Syringe-Filter with a hydrophilic polyether sulfone membrane with a 0.22 μm pore size (TPP Techno Plastic Products AG, Trasadingen, Switzerland). Moreover, this filtrated supernatant was ultrafiltrated using Amicon Ultra-15 Centrifugal Filter Devices for 45 min at 1500× *g*, 4 °C. The ultrafiltrated fraction (UFF) was suspended in 1× TBS (137 mM NaCl, 2.68 mM KCl, 25 mM Tris) and ultrafiltrated again. The obtained UFF was suspended in 1× TBS, adjusting to a final volume of 1.0 mL.

To isolate EVs from the UFF, two methods were used: polymer precipitation and affinity purification (Figure 1). The total exosome isolation reagent (Invitrogen, Waltham, MA, USA) was used for polymer precipitation according to the instruction manual. In affinity purification, the MagCapture Exosome Isolation Kit PS (FUJIFILM, Tokyo, Japan) was used. Briefly, magnetic beads bearing on Tim-4, which can bind to phosphatidylserine on EVs, were used in this procedure.

The 1.0-mL suspension of UFF was divided into 500 μL aliquot samples. Then, 500 μL of 1× TBS was added to each aliquot sample. EVs were isolated from 1.0 mL of the UFF aliquot sample by the MagCapture Exosome Isolation Kit PS according to the instruction manual. The Tim-4 affinity purification fraction (Af-F) and its supernatant (Af-S) were collected, and both were separately suspended in 1× TBS to attain a 1.0 mL final volume. The Af-F and Af-S solutions were each mixed separately with 500 µL total exosome isolation reagent and incubated overnight at 4 °C. These mixtures were centrifuged for 1 h at 10,000× *g*, 4 °C. The supernatants were removed, and the pellets were suspended in 1× TBS and collected. Another UFF aliquot was also subjected to polymer precipitation by the total exosome isolation reagent without affinity purification. The pellet obtained using the aforementioned method was also suspended in 1× TBS and collected.

### 2.3. Nano Tracking Analysis and Transmission Electron Microscopy Analysis

The nano tracking analysis (NTA) and morphological analysis of EVs through transmission electron microscopy (TEM) were outsourced to FUJIFILM Wako Bio Solutions Corporation (Fukushima, Japan). NanoSight LM10 (Malvern Panalytical, Malvern, UK) was used for the NTA. Negatively stained EV samples were prepared at the Hanaichi UltraStructure Research Institute (Okazaki, Japan). JEM-1400 Flash electron microscopy (JEOL, Tokyo, Japan) was used for TEM analysis. For negative staining, a droplet of the sample was placed on the carbon film grid for 10 s. Excess liquid was blotted off by touching the one end of the grid with a filter paper. After the grid was partially dried, a drop of the staining solution, 2% uranyl acetate into water, was added on the grid and allowed to remain on the grid for 10 s. Excess liquid was blotted off with a filter paper, and the grid was dried at room temperature.

### 2.4. Western Blotting

First, the 50 μL solutions of UFF, Af-F, and Af-S were lysed in a sample buffer (125 mM Tris-HCl, 4% SDS, 20% glycerol, 0.01% bromophenol blue, and 10 mM β-mercaptoethanol) containing the Halt Protease and Phosphatase Inhibitor Cocktail (Thermo Fisher scientific, Waltham, MA, USA). These lysates were sonicated, boiled at 98 °C for 5 min, separated on 8% SDS–polyacrylamide gels, and electroblotted onto polyvinylidene difluoride membranes (GE Healthcare, Little Chalfont, UK). The membranes were blocked for 1 h in Blocking One Solution (Nacalai Tesque, Kyoto, Japan) or 5% (*w*/*v*) non-fat milk in TBST (20 mM Tris base, 137 mM sodium chloride, 0.1% Tween 20, and pH 7.6). The blocked membranes were probed with antibodies against CD9 (mouse antibody) (Santa Cruz Biotechnology, Dallas, TX, USA) and CD81 (rabbit antibody) (Cell Signaling Technology, Danvers, MA, USA), HTLV-1 gp46 antibody (mouse antibody) (Abcam, Cambridge, UK), HTLV-1 p19 antibody (mouse antibody) (Santa Cruz Biotechnology, Dallas, TX, USA), and HTLV-1 p24 antibody (mouse antibody) (Abcam, Cambridge, UK) at 4 °C overnight. HRP-conjugated anti-rabbit or anti-mouse IgG antibodies (Cell Signaling Technology Japan, Tokyo, Japan) were used as secondary antibodies. Signals were detected using ECL western blotting detection reagents (Cytiva, Tokyo, Japan) and Fusion Solo.7S.Edge (Vilber, Collégien, France).

### 2.5. Proteome Analysis

After polymer precipitation was performed using the TEI reagent, the pellets were collected from Af-F and Af-S. Each pellet was suspended with 200 μL of 2% SDC, and trypsin buffer was added to each sample. Each mixture was incubated at 37 °C for 20 h. The supernatants were collected from each sample after centrifugation at 15,000× *g*, 4 °C for 10 min. Formic acid at a final concentration of 0.8% was added to each supernatant. These samples were centrifuged at 4 °C, 15,000× *g* for 10 min, and the supernatants were collected. The aliquot of ethyl acetate was added to the supernatant of each sample and centrifuged at 4 °C, 15,000× *g* for 10 min. Then, the pellets were collected from the bottom layer of each supernatant. Finally, each pellet was resolved with an aliquot of 0.1% formic acid. The final protein concentration was 500 ng/μL in each sample. Mass spectrometry and chromatography were performed using the Q-Exactive spectrometer (Thermo Fisher Scientific, Waltham, MA, USA) and EASY-Spray column Ultimate 3000 RSLCnano (Thermo Fisher Scientific, Waltham, MA, USA), respectively. Sequest HT of Proteome Discoverer software version 2.5 (Thermo Fisher Scientific, Waltham, MA, USA) was used for protein identification. The false discovery rate, which indicates the reliability of protein identification, was set to 1%. The peptide abundance ratio (log2) was compared between Af-F and Af-S. Proteome data can be found in jPOST repo (Japan proteome standard repository) under the ID: JPST002330 and/or http://www.proteomexchange.org/ (accessed on 24 September 2023), under the ID: PXD045623.

### 2.6. RNA Isolation

Total RNA was isolated using the TRIzol reagent (Invitrogen, Waltham, MA, USA). Briefly, 150 μL solution of UFF, Af-F, and Af-S was lysed in 1.0 mL TRIzol reagent. The isolated total RNA was treated with ezDNase (Invitrogen, Waltham, MA, USA). The total RNA concentration was measured using Nano Drop LITE (Thermo Fisher Scientific, Waltham, MA, USA).

### 2.7. Polymerase Chain Reaction (PCR)

To detect HTLV-1, *Tax*/*Rex* mRNA, and *HBZ* mRNA, reverse transcription (RT)-PCR was performed. First, 1.0 μg RNA was reverse transcribed to complementary DNA (cDNA) by a SuperScript IV First-Strand Synthesis System (Invitrogen, Waltham, MA, USA) with oligo dT primers according to the instructions. The PCR primers for *Tax*/*Rex* cDNA were as follows: forward primer (Tax-F 5′-CCCGCCGATCCCAAAGAAA-3′: positions 5169–5187) and reverse primer (Tax-R 5′-GGGTATCCGAAAAGAAGACTCTG-3′: positions 7345–7367). The PCR primers for *HBZ* cDNA were as follows: forward primer (HBZ-F 5′-GGCAGAACGCGACTCAACC-3′: positions 8728–8710) and reverse primer (HBZ-R 5′-CGGGCATGACACAGGCAAG-3′: positions 7259–7277) [18]. Based on the amount of PCR products, the PCR cycles were adjusted to 40 for *Tax*/*Rex* and *HBZ* cDNAs. PCR was performed using the Thermal Cycler Gene Atlas (ASTEC, Fukuoka, Japan). PCR products were electrophoresed on a 1.0% agarose gel. The gel was stained with ethidium bromide, which was added using an ethidium bromide dropper bottle (Genesee Scientific, San Diego, CA, USA), and visualized.

### 2.8. Long PCR

RT-long PCR was performed to detect HTLV-1 genomic RNA, including complete and defective virus genomic RNAs. First, 1.0 μg RNA was reverse transcribed to cDNA by the SuperScript IV First-Strand Synthesis System (Invitrogen, USA) with oligo dT primers according to the instructions. RT was performed with oligo dT targeting the poly A tail of HTLV-1 genomic RNA. According to previous reports, the PCR primers for HTLV-1 genomic cDNA targeting the preserved site in both complete and defective virus genomic cDNAs were as follows: forward primer (HTLV-647F 5′-GTTCCACCCCTTTCCCTTTCATTCACGACTGACTGC-3′) and reverse primer (HTLV-8345R 5′-GGCTCTAAGCCCCCGGGGGATATTTGGGGCTCATGG-3′) [19]. The PCR primers for the defective virus genomic cDNA were as follows: forward primer for both adjunctive sites of the deficient virus (positions 1333–6658) (HTLV-1318F 5′-CCAGTTTATGCAGACCATCCCTGTAAACC-3′, positions 1318–1332 and 6659–6672) and reverse primer HTLV-8345R. The PCR primers for the complete virus genomic cDNA were as follows: forward primer HTLV-647F and reverse primer (HTLV-3085R 5′-TCCATGTACTGAAGAATAGTGCATTGGGG-3′). Yeast RNA was added to the PCR reaction mixture at a final concentration of 100 ng/μL to increase the amplification efficiency. Based on the amount of PCR products, the PCR cycles were adjusted to 18, 18, and 23 for HTLV-1, defective virus, and complete virus genomic cDNAs, respectively. PCR was performed using the Thermal Cycler Gene Atlas. PCR products were electrophoresed on a 0.8% agarose gel. The gel was stained with ethidium bromide, which was added using an ethidium bromide dropper bottle, and visualized.

## 3. Results

### 3.1. Particle Size and Concentration of UFF, Af-F, and Af-S by NTA

The size and concentration of particles in UFF, Af-F, and Af-S were measured through NTA by NanoSight. The “mode” value indicates the size of the most numerous particles in each sample. Meanwhile, the “mean” value indicates the average size of all particles in each sample. The mode values of UFF, Af-F, and Af-S were 158, 133, and 180 nm, respectively (Figure 2A–C), while their mean values were 233, 190, and 283 nm, respectively (Figure 2A–C). The particle concentrations of UFF, Af-F, and Af-S were 5.93, 1.15, and 7.80 × 10^10^ particles/mL, respectively (Figure 2D).

### 3.2. Morphological Observations of EVs in UFF, Af-F, and Af-S through TEM

TEM analysis of UFF, Af-F, and Af-S samples was performed to observe the morphology of EVs and virions. According to several studies, virions or virus-like particles were defined as vesicles with a spike structure, such as those in human immunodeficiency virus type 1 (HIV-1) [20,21] and coronavirus [22]. EVs with no spike structure were observed in UFF samples (Figure 3A). The number of EVs per field of view in the TEM image was considerably lower for UFF than for Af-F (Figure 3A,B). In Af-F, many EVs sized ~30–130 nm were observed and particles with spike structures that could be considered virions or virus-like particles were not present (Figure 3B,D,E). Observing EVs or virus-like particles in one field of view of the TEM image was difficult for Af-S because of the high level of impurities (Figure 3C).

### 3.3. Existence of CD9, CD81, and HTLV-1 Virion Proteins in Tim-4 Affinity Immunoprecipitation Samples of EVs

To detect proteins of EVs and HTLV-1 virions, western blotting was performed using UFF, Af-F, and Af-S (Figure 4). In UFF, both EV markers and HTLV-1 virion proteins were observed, namely CD9, CD81, gp46, p19, and p24. Meanwhile, both CD9 and CD81, but not the HTLV-1 envelope glycoprotein gp46, were detected in Af-F. gp46 was detected in Af-S. The HTLV-1 matrix protein p19 and capsid protein p24 were detected in all of the three samples. The amount of p19 and p24 appeared to be greater in Af-F than in Af-S.

### 3.4. Proteome Analysis of the EV Fraction Obtained through Tim-4 Affinity Immunoprecipitation

Proteome analysis of the EV fraction obtained through Tim-4 affinity immunoprecipitation was performed to evaluate the purity of EVs in the Af-F and Af-S fractions. Figure 5 shows the representative profiles of proteins in Af-F and Af-S determined through LC-MS/MS analysis. The graph displays the Af-F/Af-S abundance ratio (log2) of EV markers and ribosomal proteins. Ribosomal proteins are recommended as a purity control for EVs in the MISEV2018 guidelines [11]. The EV protein markers in Af-F were more than those in Af-S, such as CD9, CD81, HLA, LAMP, and annexin (Figure 5A). On the other hand, HSPA8, HSP90AB1, actin, tubulin, and GAPDH were more in Af-S than in Af-F (Figure 5A). Most ribosomal proteins were enriched in AF-S compared with Af-F (Figure 5B). Therefore, proteomic analysis confirmed that the combination of the Tim-4 affinity method and polymer precipitation method can collect and purify EVs from the HTLV-1 cell culture medium.

### 3.5. EVs Derived from HTLV-1-Infected Cell Line Contain Tax/Rex mRNA and HTLV-1 Genome RNA

RT-PCR was performed to determine HTLV-1 *Tax*/*Rex* and *HBZ* mRNAs in UFF, Af-F, and Af-S samples. Long PCR was performed to detect HTLV-1 virus genomic RNAs, namely defective and complete virus genomic RNAs. Both RT-PCR and long PCR products were electrophoresed on a 1.0% or 0.8% agarose gel and visualized with ethidium bromide. *Tax*/*Rex* mRNA, but not *HBZ* mRNA, was detectable in UFF, Af-F, and Af-S (Figure 6A,B). Figure 6C presents a schematic diagram of the structure of HTLV-1 genomic RNA and the location of long PCR primers. Long PCR was performed to detect both defective and complete virus genomic RNAs by the forward primer HTLV-647F and the reverse primer HTLV-8345R. Complete virus genomic RNAs were observed as 7.7-kB PCR products in the positive control. Defective virus genomic RNAs with a 5.3-kB deletion (positions 1333–6658) were observed as 2.4-kB PCR products. These PCR products of the defective RNAs (2.4 kB) were detectable in UFF, Af-F, and Af-S. On the other hand, the PCR products of complete RNAs (7.7 kB) were undetectable in these samples (Figure 6D). Long PCR was performed to detect only defective RNAs by the forward primer HTLV-1318F and the reverse primer HTLV-8345R. Defective virus genomic RNAs were observed as 1.7-kB PCR products in the positive control. These PCR products of defective RNAs (1.7 kB) were detectable in UFF, Af-F, and Af-S (Figure 6E). Long PCR was performed to detect only complete virus genomic RNAs by the forward primer HTLV-647F and the reverse primer HTLV-3085R. Complete virus genomic RNAs were observed as 2.4-kB PCR products in the positive control. These PCR products of the complete RNAs (2.4 kB) were detectable in UFF, Af-F, and Af-S (Figure 6F). Sequence analysis confirmed that PCR products in Af-F (Figure 6D–F) were HTLV-1 virus genomic RNA (Appendix A Appendix A).

## 4. Discussion

Here, we investigated a method for separating and collecting EVs derived from HTLV-1-infected cells from the culture supernatant. The Tim-4 affinity immunoprecipitation method, which targets phosphatidylserine exposed on the EV membrane, was considered useful for separating and collecting EVs and HTLV-1 virus particles. Furthermore, EVs were easily concentrated and purified by combining polymer precipitation and Tim-4 affinity immunoprecipitation. EVs collected from the culture supernatant of the HTLV-1-infected cells were suspected to contain HTLV-1 genomic RNA.

The HTLV-1-infected cell line MT-2 produces virus particles with a diameter of approximately 100 nm [23,24]. Therefore, HTLV-1 virus particles and EVs were considered to have similar particle sizes. Separating virus particles by EV purification and collection methods based on the particle size or relative density, such as ultracentrifugation, density gradient centrifugation, and polymer precipitation, is difficult [8,25,26]. Furthermore, a study in which EVs derived from HTLV-1-infected cells were collected using density gradient centrifugation suggested that a portion of the collected fraction may contain virus particles [27]. In this study, first, the cell culture medium from which cells, large-sized microvesicles, and apoptotic bodies were removed was subjected to gravity gradient centrifugation and filtration to prepare fractions enriched with virus particles and EVs. Furthermore, whether EVs and HTLV-1 virus particles can be separated through immunoprecipitation by Tim-4-bound magnetic beads that bind to phosphatidylserine exposed on the EV membrane was examined. Immunoprecipitation targeting CD9, CD63, and CD81 exposed on the EV membrane is a specific collection method targeting the membrane. Collection methods targeting these EV marker proteins cause EV damage during EV elution. Therefore, to analyze the biological functions of EVs, a collection method that causes less damage to EVs must be used [8,26]. On the other hand, Tim-4 affinity immunoprecipitation can collect intact EVs [28].

NTA allows nanoparticles of sizes 20–1000 nm in liquid suspension samples to be directly and individually visualized and counted in real time. Simultaneously, NTA provides high-resolution particle size distribution profiles and concentration measurements. Therefore, it is possible to evaluate the size of all particles present in liquid samples by NTA. In this study, NTA was used to obtain the particle size distribution profiles and concentration of particles in UFF, Af-F, and Af-S. According to the mode value, the most common particle sizes were approximately 158, 138, and 180 nm in UFF, Af-F, and Af-S, respectively (Figure 2). Conversely, TEM analysis revealed the presence of EVs measuring approximately 30–130 nm in the Tim-4 affinity-purified fraction Af-F (Figure 3B,D,E). The particle size observed by these analyses differed. One limitation of TEM is that not all particles in the prepared sample can be observed, and it is difficult to evaluate the size and morphology of all particles. Additionally, impurities can hinder the observation of particles, as observed in case of Af-S. As it is difficult to observe EVs using TEM, NTA is considered an appropriate method to analyze the particle size distribution and concentration of EVs in liquid samples.

We performed TEM to directly confirm EVs or virus particles in the samples collected using our method. It is generally reported that EVs do not possess the spike structure found in viruses. For example, EVs produced by influenza virus-infected cells do not have the spike structure found in influenza viruses [29]. Therefore, we considered it important to confirm the presence of spike structures to distinguish EVs from HTLV-1 particles. Furthermore, because the spike structure of HTLV-1 particles is difficult to observe via TEM using thin sections [24], negative-stain TEM was performed without using thin sections in this study. No virus-like particles with a spike structure were observed in the Tim-4 affinity-purified fraction through TEM, and many EVs with a relatively uniform particle size were observed (Figure 3B,D,E). On the other hand, the collected Af-S contained many contaminants, which made observing particle structures of virus-like particles difficult (Figure 3C). Protein analysis of each collected fraction revealed EV markers (CD9 and CD81), but not the HTLV-1 viral envelope protein gp46, in Af-F. Because gp46 was confirmed in Af-S obtained through Tim-4 affinity immunoprecipitation, most virus particles were not pulled down by this method and were present in Af-S. Although, given that gp46 is often shed from viral particles it would be important to address this possibility of the presence of gp46-negative viruses in the Af-F fraction, it is unlikely that all gp46 was lacking from the virus particle. If HTLV-1 particles partially lacking gp46 were collected using the Tim-4 affinity method, gp46 should have been detected in the Af-F fraction. However, gp46 was not detected in the Af-F fraction and gp46 was detected in the Af-S fraction at the same level as UFF. Therefore, in this study, it was thought that EVs other than virus particles were collected in the Af-F fraction, but it could not be denied that virus particles that completely lack gp46 exist in the Af-F fraction. Because p19 and p24 were detected not only in Af-S but also in Af-F, EVs were considered to possibly contain p19 and p24 (Figure 4).

Exosome formation involves the following steps: formation of intraluminal vesicles (ILVs) composed of endosomal membranes, fusion of the ILV-containing multivesicular body with the cell membrane, and release of ILVs themselves outside the cell as exosomes. Microvesicles are also formed through direct budding of the cell membrane outside the cell. Proteins that compose exosomes are selectively incorporated into the exosomes when forming ILVs. Proteins that compose microvesicles are also selectively incorporated during budding [2]. EVs collected through the Tim-4 affinity method may have been loaded with p19 and p24 as well as CD9 and CD81. Therefore, some of the EVs collected through the Tim-4 affinity method may contain p19 and p24 during EV formation and be released extracellularly. A previous study by Jaworski et al. indicated that exosomes purified from MT-2 cell culture supernatants might contain p24 protein [16]. However, exosome purification methods, such as the NT particle method, might not have sufficient accuracy to avoid contamination by HTLV-1 virions; thus, it remains unclear whether EVs derived from HTLV-1-infected cells contain p24 [16]. The present study indicated that EVs derived from HTLV-1-infected MT-2 cells collected by Tim-4 affinity immunoprecipitation contained HTLV-1-related proteins such as p19 and p24. In addition, EVs derived from HTLV-1 Gag-expressing Jurkat cells have been reported to contain the HTLV-1 Gag protein [30]. It has been also reported that EVs derived from HTLV-1-infected cells contain p19 [31]. Another study demonstrated that Evs derived from Bovin leukemia virus-infected cells contain p24 [32]. Considering previous reports and our study, Evs derived from HTLV-1-infected cells might contain viral structural proteins such as capsid and matrix proteins during EV formation.

The Tim-4 affinity immunoprecipitation method is based on the binding of Tim-4 to phosphatidylserine exposed on the constituent EV membrane. In this study, gp46, the HTLV-1 virus envelope protein, was not detected in the Tim-4 affinity fractions (Af-F). Therefore, we inferred that almost no phosphatidylserine was present in the HTLV-1 virion envelope. The particles of HIV-1, a retrovirus similar to HTLV-1, have exposed phosphatidylserine, which is crucial for HIV to infect cells [33]. In the HIV mode of transmission, HIV virions directly infect lymphocytes. Conversely, HTLV-1 requires cell-to-cell contact via virological synapses and infection by free HTLV-1 virions is believed to rarely occur [34,35]. The localization of phosphatidylserine in HTLV-1 particles is unknown. The absence of the HTLV-1 envelope protein in the EV fraction collected using the Tim-4 affinity method in this study suggested that HTLV-1 particles have almost no phosphatidylserine.

Proteomic analysis of EV fractions collected using the Tim-4 affinity method revealed that the Tim-4 affinity fractions contained more EV marker proteins and less ribosomal proteins (Figure 5). The fractions collected using the Tim-4 affinity method were considered highly purified fractions of Evs. However, HSPA8, HSP90AB1, actin, tubulin, and GAPDH, which have been reported as EV marker proteins, were found more frequently in Af-S than in Af-F (Figure 5). Exomeres, vesicles smaller than exosomes, have recently been identified [36]. Exomeres contain less phosphatidylserine than Evs and abundant HSPA8, HSP90AB1, actin, tubulin, and GAPDH [11,37]. Therefore, in this affinity immunoprecipitation method, exomeres may remain in Af-S.

Evs derived from HTLV-1-infected cell lines contain HBZ and Tax/Rex mRNAs [15,16]. In this study, Evs collected using the Tim-4 affinity method contained Tax/Rex mRNA, but HBZ mRNA was not detected (Figure 6A,B). Because HBZ mRNA is localized in the nucleus [38,39] and its abundance in the cytoplasm is low, we assumed that uptake of HBZ mRNA into Evs is low. In addition, the gene expression level of HBZ mRNA in the MT-2 cells used in this study is reported to be lower than that of Tax mRNA [18]. The condition of gene expression may affect the content of these mRNAs in Evs.

In this study, defective and complete HTLV-1 genomic RNAs were detected in both Af-F collected using the affinity method and their supernatant fractions, Af-S (Figure 6D–F). The main HTLV-1 proviruses of MT-2 are the full-length provirus and the defective provirus lacking the 1333–6658 region [18]. HTLV-1 genomic RNA is transcribed from the provirus and incorporated into virus particles. Because Evs are loaded with cell-derived nucleic acids and proteins [2], they may contain viral genomic nucleic acids and virus-associated proteins. In fact, Evs derived from HCV-infected cells contain HCV genomic RNA [12]. The interaction between the Gag protein and genomic RNA is crucial for HTLV-1 virion formation. For example, p15 interacts with the psi element of HTLV-1 genomic RNA and p19 also interacts with HTLV-1 genomic RNA [40]. P24 is important for Gag protein formation and Gag–Gag interactions [41]. Therefore, when either the Gag protein or HTLV-1 genomic RNA is loaded into Evs, other substances might also be taken up with them. In addition, the Gag protein in Evs might protect virus genomic RNA in the same way as that observed in virions. In this study, EVs collected using the Tim-4 affinity method were speculated to incorporate HTLV-1 genomic RNA, p19, and p24 into EVs. Moreover, EVs collected using the affinity method did not contain the envelope protein gp46. These EVs may be not recognized by the anti-HTLV-1 gp46 antibody, a neutralizing antibody of HTLV-1. On the other hand, because EVs collected using the Tim-4 affinity method are rich in phosphatidylserine, these particles are likely to be taken up into cells through the scavenger receptor or through micropinocytosis [42,43]. A previous report suggests that EVs derived from HTLV-1-infected cells assist cell-to-cell contact and promote HTLV-1 virus infection [27]. Although the phosphatidylserine-rich EVs that were isolated and purified functioned as transport carriers for the viral genome, further detailed studies investigating the function of EVs in the HTLV-1 infection mechanism are warranted.

This study has several limitations. First, electron microscopic observation could not directly confirm whether virus particles could be removed by our method. Observing virus particles was difficult because many impurities were present in samples collected using the ultrafiltration membrane. Visualizing the spike structure of HTLV-1 particles under an electron microscope was also difficult [24], and distinguishing EVs from virus particles based on their morphology as observed under an electron microscope might not be possible. Therefore, in the present study, we performed both TEM and immunoblotting and confirmed the expression of HTLV-1-related proteins in Tim-4 affinity IP samples and Af-S samples. Second, because gp46 is often shed from virus particles, we cannot rule out the possibility that there are virus particles in which gp46 has been completely shed in the fractions collected using the Tim-4 affinity method. Third, whether the proposed method is suitable for collecting EVs from HTLV-1-infected cell lines other than MT-2 cells remains unclear and this needs to be elucidated in future studies. Fourth, EVs collected using the Tim-4 affinity method were those with exposed phosphatidylserine, and other EVs were not collected. In the future, when evaluating the physiological activity of EVs collected using this method, one should be aware that the characteristics of all EVs are not captured. Additionally, because EVs collected using our method contain HTLV-1-related proteins and HTLV-1 genomic RNA, it is necessary to investigate the infectivity of EVs purified by our method in the future. It would be extremely interesting to clarify whether EVs serve as HTLV-1 infection vectors. Such studies could lead to the discovery of a novel transmission mechanism of HTLV-1 via EVs derived from HTLV-1-infected cells.

## 5. Conclusions

We proposed a method for purifying EVs from the culture supernatant of the HTLV-1-infected cell line MT-2 while limiting contamination with virus particles. EVs purified using this method may contain the virion core protein p24 and the HTLV-1 viral genome. This method may be useful for investigating the biological characteristics and functional analysis of EVs derived from HTLV-1-infected cells.

## Figures and Tables

**Figure 1 viruses-16-00249-f001:**
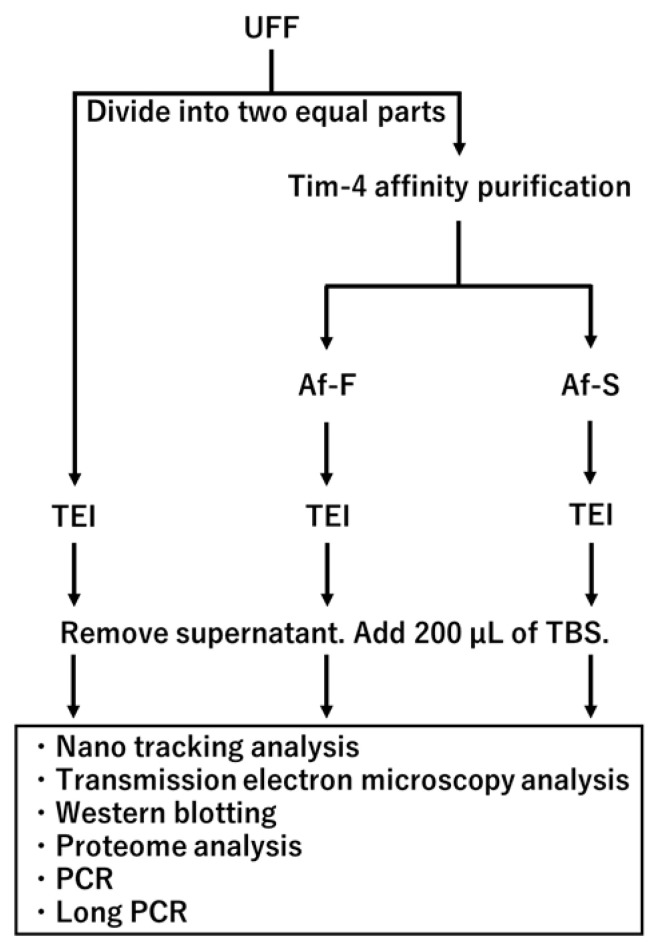
Flow diagram depicting sample preparation of extracellular vesicles derived from HTLV-1-infected cell line. The ultrafiltrated fraction (UFF) was divided into two equal parts: one for polymer precipitation using a solution of total exosome isolation (TEI) and the other one for Tim-4 affinity purification. The Tim-4 affinity purification fraction (Af-F) and its supernatant fraction (Af-S) were subjected to polymer precipitation using TEI. Each polymer precipitation fraction was resuspended with TBS and analyzed separately.

**Figure 2 viruses-16-00249-f002:**
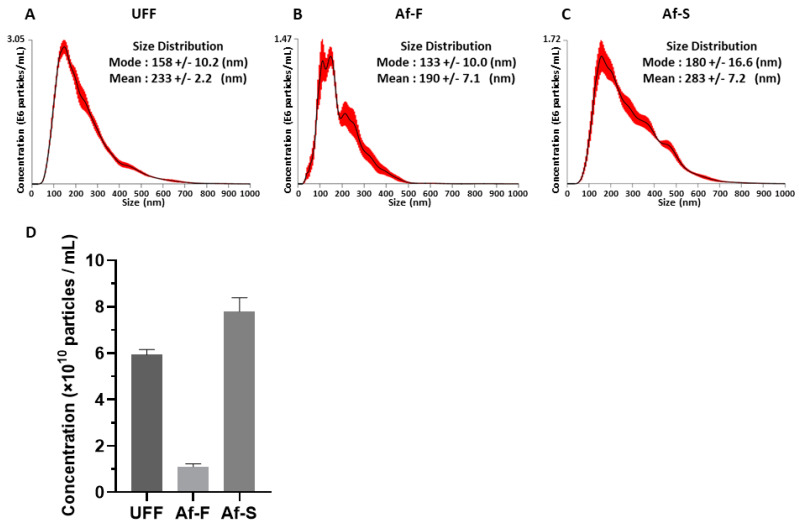
Analysis of extracellular vesicles’ size and concentration. The ultrafiltrated fraction (UFF), Tim-4 affinity purification fraction (Af-F), and its supernatant fraction (Af-S) were analyzed by NanoSight (**A**–**C**). Histograms show the average value of the microvesicles’ size of 5 measurements for each sample. Red lines indicate mean ± standard error of the mean (SEM). The concentration of microvesicles in UFF, Af-F, and Af-S by NanoSight analysis (**D**). This graph depicts the mean (±SEM) of the results of 5 measurements for each sample.

**Figure 3 viruses-16-00249-f003:**
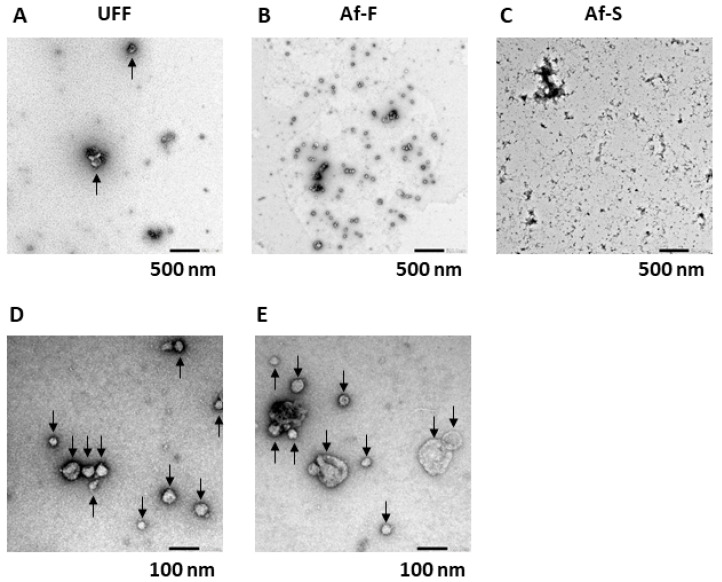
Morphological observation of extracellular vesicles using transmission electron microscopy. The ultrafiltrated fraction (UFF), Tim-4 affinity purification fraction (Af-F), and its supernatant fraction (Af-S) were analyzed using transmission electron microscope (**A**–**C**). Magnification, ×10 k. Scale bar = 500 nm. Af-F was observed using TEM (**D**,**E**). Magnification, ×50 k. Scale bar = 100 nm. Arrows indicate EVs.

**Figure 4 viruses-16-00249-f004:**
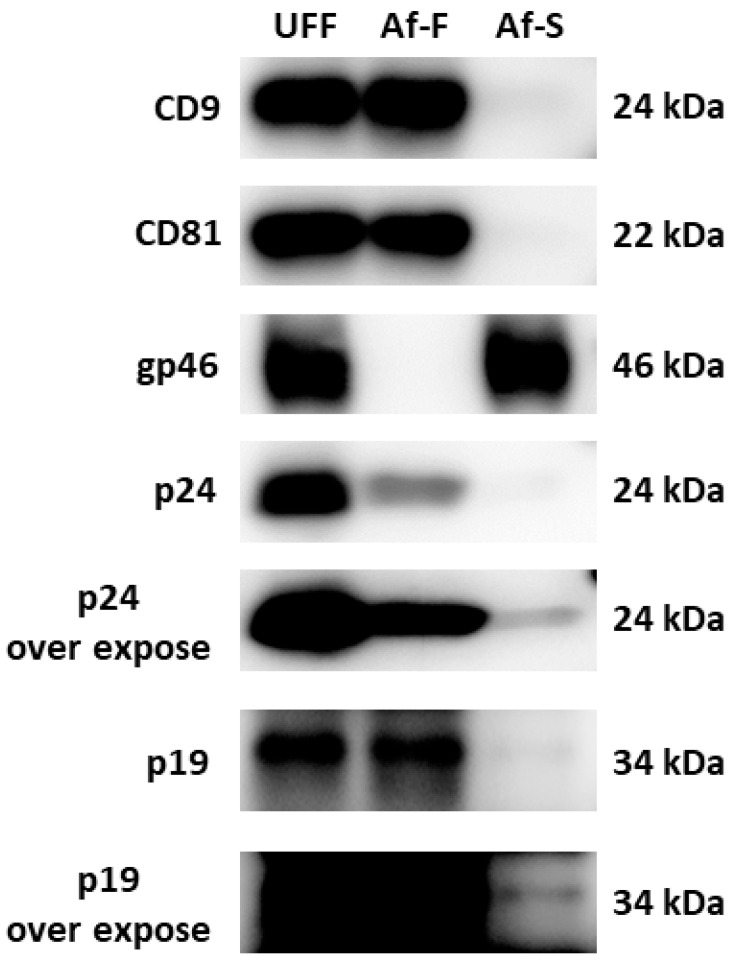
Immunoblotting for extracellular vesicles (EVs) component proteins and human T-cell leukemia virus type 1 (HTLV-1) virion proteins in ultrafiltrated fraction (UFF), Tim-4 affinity-purified fraction (Af-F), and supernatant fraction (Af-S). Both CD9 and CD81 are well known as component proteins of EVs. Glycoprotein 46 (gp46) is one of the envelope proteins, p19 is a matrix protein, and p24 is a capsid protein of HTLV-1.

**Figure 5 viruses-16-00249-f005:**
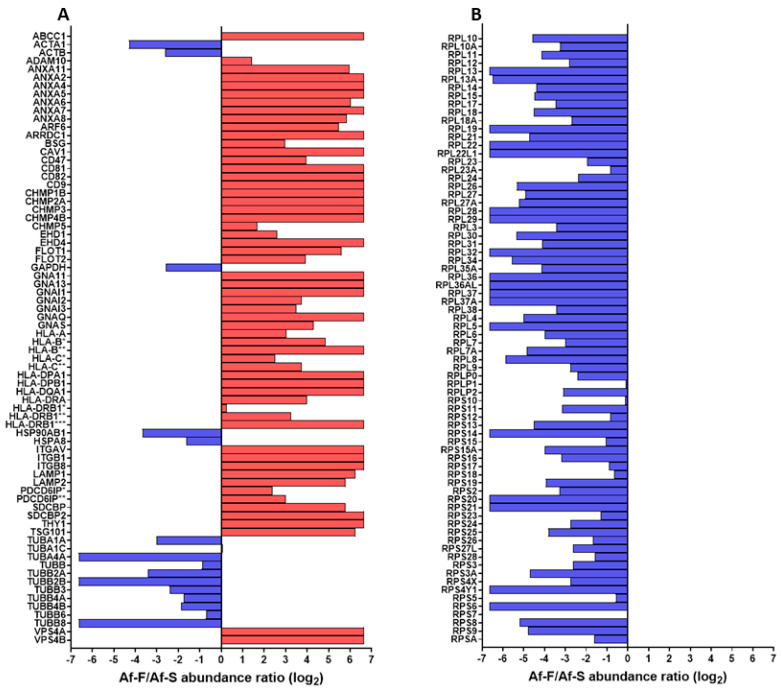
Abundance ratio (log2) of component proteins of extracellular vesicles (EVs) and ribosomal proteins in Tim-4 affinity purification fraction (Af-F) and its supernatant fraction (Af-S). Af-F/Af-S abundance ratio (log2) is shown as a red bar when it is greater than 0, and a blue bar when it is less than 0. (**A**) Af-F/Af-S abundance ratio (log2) of EVs’ component proteins. The name of each protein is indicated by a gene symbol. ABCC1; Multidrug resistance-associated protein 1. ACTA1; Actin, alpha skeletal muscle. ACTB; Actin, cytoplasmic 1. ADAM10; Disintegrin and metalloproteinase domain-containing protein 10. ANXA; Annexin A. ARF6; ADP-ribosylation factor 6. ARRDC1; Arrestin domain-containing protein 1. BSG; Basigin. CAV1; Caveolin-1. CD; CD antigen. CHMP; Charged multivesicular body protein. EHD; EH domain-containing protein. FLOT; Flotillin. GAPDH; Glyceraldehyde-3-phosphate dehydrogenase. GNA; Guanine nucleotide-binding protein subunit alpha. GNAI; Guanine nucleotide-binding protein G(i) subunit alpha. GNAQ; Guanine nucleotide-binding protein G(q) subunit alpha. GNAS; Guanine nucleotide-binding protein G(s) subunit alpha isoforms XLas. HLA-A; HLA class I histocompatibility antigen, A-24 alpha chain. HLA-B*; HLA class I histocompatibility antigen, B-41 alpha chain. HLA-B**; HLA class I histocompatibility antigen, B-52 alpha chain. HLA-C*; HLA class I histocompatibility antigen, Cw-2 alpha chain. HLA-C**; HLA class I histocompatibility antigen, Cw-3 alpha chain. HLA-DPA1; HLA class II histocompatibility antigen, DP alpha 1 chain. HLA-DPB1; HLA class II histocompatibility antigen, DP beta 1 chain. HLA-DQA1; HLA class II histocompatibility antigen, DQ alpha 1 chain. HLA-DRA; HLA class II histocompatibility antigen, DR alpha chain. HLA-DRB1*; HLA class II histocompatibility antigen, DRB1 beta chain. HLA-DRB1**; HLA class II histocompatibility antigen, DRB1-16 beta chain. HLA-DRB1***; HLA class II histocompatibility antigen, DRB1-9 beta chain. HSP90AB1; Heat shock protein HSP 90-beta. HSPA8; Heat shock cognate 71 kDa protein. ITG; Integrin. LAMP; Lysosome-associated membrane glycoprotein. PDCD6IP*; Isoform 2 of Programmed cell death 6-interacting protein. PDCD6IP**; Programmed cell death 6-interacting protein. SDCBP; Syntenin-1. SDCBP2; Syntenin-2. THY1; Thy-1 membrane glycoprotein. TSG101; Tumor susceptibility gene 101 protein. TUB; Tubulin. VPS; Vacuolar protein sorting-associated protein. (**B**) Af-F/Af-S abundance ratio (log2) of ribosomal proteins. The name of each protein is indicated by a gene symbol. RPL; 60S ribosomal protein L. RPLP; 60S acidic ribosomal protein P. RPS; 40S ribosomal protein S.

**Figure 6 viruses-16-00249-f006:**
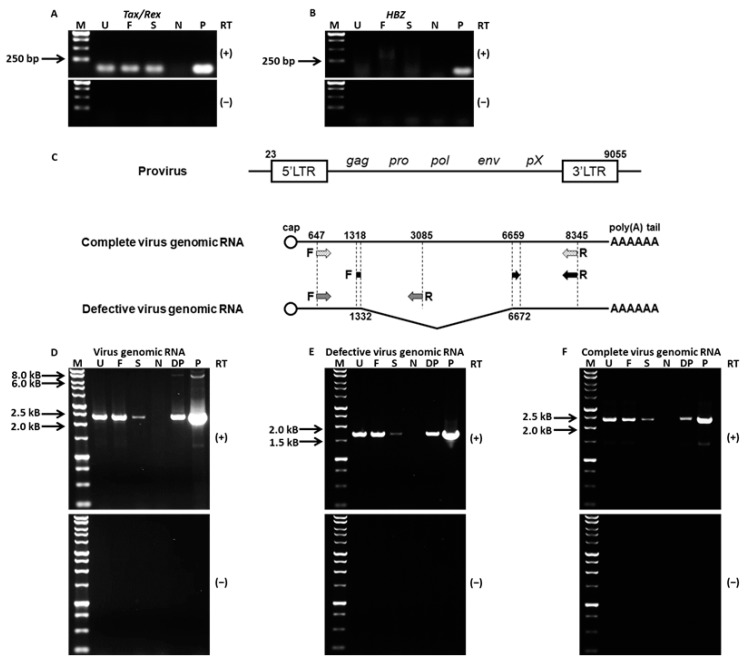
Extracellular vesicles derived from MT-2 contain not only genomic RNA of human T-cell leukemia virus type 1 (HTLV-1) but also mRNA coding *Tax*/*Rex* and *HBZ*. RNA was extracted from the ultrafiltrated fraction (UFF), Tim-4 affinity purification fraction (Af-F), and its supernatant fraction (Af-S). Reverse transcription was performed by oligo dT targeting poly A tails of HTLV-1-related mRNA and HTLV-1 genomic RNA and detected by PCR and long PCR, respectively (*n* = 3). (**A**,**B**) PCR analysis of HTLV-1 *Tax*/*Rex* and *HBZ* mRNA in UFF, Af-F, and Af-S. RT (+): reverse transcribed, RT (−): not reverse transcribed. M: 100 bp ladder marker, U: UFF, F: Af-F, S: Af-S, N: negative control (Jurkat cells), P: positive control (MT-2 cells). (**C**) Schematic diagram showing the structure of HTLV-1 genomic RNA and the locations of long PCR primers. The sequence numbers of genomic RNA are indicated as HTLV-1 provirus base numbers (NCBI accession No. J02029). LTR: long terminal repeat, F: forward primer, R: reverse primer. (**D**–**F**) Long PCR analysis of defective HTLV-1 genomic RNA and full-length HTLV-1 genomic RNA in UFF, Af-F, and Af-S with HTLV-647F and HTLV-8345R (**D**), HTLV-1318F and HTLV-8345R (**E**), and HTLV-647F and HTLV-3085R (**F**). RT (+): reverse transcribed, RT (−): no reverse transcribed. M: 100 bp ladder marker, U: UFF, F: Af-F, S: Af-S, N: Negative control (Jurkat cell), P: Positive control (MT-2 cell), DP: Positive control with 5-fold dilution of PCR product.

## Data Availability

Proteome data can be found in jPOST repo (Japan proteome standard repository) under the ID: JPST002330 and/or http://www.proteomexchange.org/ (accessed on 24 September 2023), under the ID: PXD045623. Other data are contained within this article or the Appendix A.

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
