# Peer review of "Purification Method of Extracellular Vesicles Derived from Human T-Cell Leukemia Virus Type 1-Infected Cells without Virions"

_viruses, 2024, doi:10.3390/v16020249_

Round 1

Reviewer 1 Report

Comments and Suggestions for Authors

Even though this method was previously reported by Nakai et al, the research reported here highlights its application to the purification of exosomes from HTLV-1 infected cells. The presence of high levels of p24 in the puried fraction is puzzling.It would be good if the authors could say whether previous studies have reported HTLV-1 p24 in exosomes before? Would it be possible that VLP consisting of p24 were also purified using this method?

- Please include arrows in TEM figures to highlight structures discussed 

Author Response

Response to Reviewer’s comments.

Dear Editor and Reviewers,

We thank you for your careful review and your pertinent and insightful comments. We have revised our manuscript following the suggestions of reviewer. New references have been added in the revised version, and the reference numbering has been updated accordingly.

The revised sections have been highlighted using yellow lines. We hope that our edits and the responses provided below satisfactorily address all issues and concerns raised by you and the reviewers.

Best regards,

Kunihiko Umekita

  • Reviewer 1

Even though this method was previously reported by Nakai et al, the research reported here highlights its application to the purification of exosomes from HTLV-1 infected cells. The presence of high levels of p24 in the puried fraction is puzzling. It would be good if the authors could say whether previous studies have reported HTLV-1 p24 in exosomes before?

Response) We appreciate your comment. A previous study by Jaworski et al. indicated that exosomes purified from MT-2 cell culture supernatant contain p24 protein (J Biol Chem. 2014 Aug 8; 289(32): 22284-305). However, exosome purification methods, such as the NT particle method, might lack accuracy to exclude HTLV-1 virion contamination. Moreover, it remains unclear whether EVs derived from HTLV-1-infected cells contain p24. Our study indicated that EVs derived from HTLV-1-infected MT-2 cells collected via Tim-4 affinity immunoprecipitation contained HTLV-1-related proteins, such as p19 and p24. In addition, EVs derived from HTLV-1 gag-expressing Jurkat cells contain HTLV-1 Gag protein (PLoS Biol. 2007;5(6):e158). It has been also reported that EVs derived from HTLV-1-infected cells contain p19 (Retrovirology. 2021;18(1):6). Another study demonstrated that EVs derived from Bovin leukemia virus-infected cells contain p24 (PLoS One. 2013 Oct 17;8(10):e77359). Therefore, EVs derived from HTLV-1-infected cells might contain viral structural proteins such as capsid and matrix proteins during EV formation. We described these points in the revised text (page 13, lines 418–430).

Would it be possible that VLP consisting of p24 were also purified using this method?

Response) Thank you for your insightful query. It is unclear whether virus-like particles (VLPs) can be purified using this method. We believe it is extremely important to discuss the differences between VLPs and extracellular vesicles (EVs). The most important discussion point is the definition of VLPs. Although our results indicated that EVs purified using our method contain viral structural proteins such as p24 and p19, they were not covered with the virus envelope protein gp46. Therefore, future investigation is warranted to clarify whether EVs lacking a viral envelope but containing matrix proteins can be considered VLPs.

- Please include arrows in TEM figures to highlight structures discussed

Response) We are grateful for your suggestion. We have added arrowheads in Figure 3 to indicate EVs, and the legend of this figure was revised accordingly.

Reviewer 2 Report

Comments and Suggestions for Authors

In this paper written by Kawano et al, authors investigated methods to purify EVs released from the HTLV-1 infected MT-2 cells, and characterized the proteins and RNA in the EVs. The data demonstrated that the EVs purified by immunoprecipitation with Tim-4 contained EV marker proteins, nucleocapsid p24 and the HTLV-1 viral genome but not gp46. Although the experiment has some minor limitations that were described in Discussion (e.g., the method is not tested with the other HTLV-1 cell lines), the data provided interesting observations and opinions in this field, and they fit well to the scope of the journal. The results are reasonable largely and support main conclusions. However, in my opinion, some points need to be addressed. I will recommend revision with a moderate modification. Some points to consider in subsequent versions are shown below.

1. Figure 2 1. showed that the EVs in Af-S had bigger and more than those in Af-F and UFF. Authors mentioned that “In Af-S, observing EV-like particles or virus-like particles in one field of view of the TEM was difficult (Figure 3C).” What are the definitions of EV-like or virus-like particles in the picture? I feel that there is a gap between the mode values of UFF, Af-F, and Af-S (each 158, 133, and 180 nm) and the TEM pictures. Authors could provide the connection between Fig. 2 and Fig.3 and the interpretation to avoid misunderstanding.

2. Authors could provide the information regarding the correlation between p24 and HTLV-1 genomic RNA, especially for the sequence containing psi in EVs.

3.     I suggest adding a diagram to show primers, PCR products, full-length viral RNA and defective viral RNA.

4.     Page 10: EVs similar in size to exosomes and microvesicles were collected. No virus-like particles with a spike structure were observed in the Tim-4 affinity-purified fraction through TEM, and many EVs with a relatively uniform particle size were observed. Does this statement correspond to Fig. 3B? If not, please provide the data (could be supplemental Fig.).

5.     I believe that the words “supernatant” in page 10 indicate Af-S. You would unify the style.

6.     Page 10: NAT revealed EV-like particles of around 30–120 nm in the Tim-4 affinity-purified fraction (Figure 3B, D, and E). Please clarify NAT?

7.     Page 10: Because gp46 was confirmed in the supernatant fraction obtained through Tim-4 affinity immunoprecipitation, most virus particles were not pulled down by this method and were present in the supernatant fraction. I suggest conducting infectivity assay with the EVs in UFF, Af-S and Af-F to test physiological conditions of the EVs.

Comments on the Quality of English Language

A minor modification is required.

Author Response

Response to Reviewer’s comments.

Dear Editor and Reviewers,

We thank you for your careful review and your pertinent and insightful comments. We have revised our manuscript following the suggestions of reviewer. New references have been added in the revised version, and the reference numbering has been updated accordingly.

The revised sections have been highlighted using yellow lines. We hope that our edits and the responses provided below satisfactorily address all issues and concerns raised by you and the reviewers.

Best regards,

Kunihiko Umekita

  • Reviewer 2

In this paper written by Kawano et al, authors investigated methods to purify EVs released from the HTLV-1 infected MT-2 cells, and characterized the proteins and RNA in the EVs. The data demonstrated that the EVs purified by immunoprecipitation with Tim-4 contained EV marker proteins, nucleocapsid p24 and the HTLV-1 viral genome but not gp46. Although the experiment has some minor limitations that were described in Discussion (e.g., the method is not tested with the other HTLV-1 cell lines), the data provided interesting observations and opinions in this field, and they fit well to the scope of the journal. The results are reasonable largely and support main conclusions. However, in my opinion, some points need to be addressed. I will recommend revision with a moderate modification. Some points to consider in subsequent versions are shown below.

  1. Figure 2 1. showed that the EVs in Af-S had bigger and more than those in Af-F and UFF. Authors mentioned that “In Af-S, observing EV-like particles or virus-like particles in one field of view of the TEM was difficult (Figure 3C).” What are the definitions of EV-like or virus-like particles in the picture? I feel that there is a gap between the mode values of UFF, Af-F, and Af-S (each 158, 133, and 180 nm) and the TEM pictures. Authors could provide the connection between Fig. 2 and Fig.3 and the interpretation to avoid misunderstanding.

Response) We appreciate your insightful comments. We wanted to directly observe the morphology of EVs and virions or virus-like particles (VLPs) using TEM. Several studies reported virions or VLPs as vesicles with a spike structure, as seen in the TEM images of human immunodeficiency virus type 1 and coronavirus under TEM (J Gen Virol. 1988;69:2455-2469; J Virol. 2017;91:e00415-17; Med J Aust. 2020;212:459-462). Conversely, we expected that EVs would appear as vesicles without spike structure. However, we could not observe any vesicles with spikes that could be considered virions or virus-like particles in UFF, Af-F, and Af-S samples. We described these points in the revised manuscript (page 6, lines 229–238).

We are grateful for your comment regarding the discrepancy in particle sizes observed using nanoparticle tracking analysis (NTA) and TEM analysis. NTA allows nanoparticles of sizes 20–1000 nm in a liquid suspension samples to be directly and individually visualized and counted in real time. Simultaneously, NTA provides high-resolution particle size distribution profiles and concentration measurements. Therefore, it is possible to evaluate the size of all particles present in liquid samples using NTA. Meanwhile, one limitation of TEM is that not all particles in the prepared sample can be observed, and it is difficult to evaluate the size and morphology of all particles. In addition, particle observation can be hampered by impurities in samples, as noted in Af-S samples. Even if it is difficult to observe EVs using TEM, NTA is considered an appropriate method to analyze the particle size of EVs present in liquid samples.

We believe that discrepancy in particle sizes obtained using NTA and TEM can be attributed to the aforementioned reasons. We have mentioned this point in the discussion and limitations sections (page 12, lines 368–382).

  1. Authors could provide the information regarding the correlation between p24 and HTLV-1 genomic RNA, especially for the sequence containing psi in EVs.

Response) We thank you for your suggestion. We have described the correlation between p24 and HTLV-1 genomic RNA, especially for the psi element of HTLV-1 genomic RNA, as follows (page 14 lines 466–472): “The interaction between the Gag protein and genomic RNA is crucial for HTLV-1 virion formation. For example, p15 interacts with the psi element of HTLV-1 genomic RNA, and p19 also interacts with HTLV-1 genomic RNA (Semin Cell Dev Biol 2019, 86:129-139.). Capsid protein, p24 is important for Gag protein formation and Gag–Gag interactions (Front Microbiol. 2014 Jun 24;5:302.). Therefore, when either the Gag protein or HTLV-1 genomic RNA is loaded into EVs, other substances might also be taken up with them. In addition, the Gag protein in EVs might protect virus-genomic RNA in the same way as observed in virions.”

  1. I suggest adding a diagram to show primers, PCR products, full-length viral RNA and defective viral RNA.

Response) We appreciate your suggestion. We have revised Figure 6. Figure 6C presents a schema of the location of the long PCR primer set as well as complete (full-length) and defective viral RNA. The legend of Figure 6 was also revised.

In addition, we have corrected the primer set names in Figure 6C (page 11, line 337–338; Figure 6 legend). The Materials and Methods (page 5, line 201) and Results (page 10, line 316) were also revised.

  1. Page 10: EVs similar in size to exosomes and microvesicles were collected. No virus-like particles with a spike structure were observed in the Tim-4 affinity-purified fraction through TEM, and many EVs with a relatively uniform particle size were observed. Does this statement correspond to Fig. 3B? If not, please provide the data (could be supplemental Fig.).

Response) We thank you for your comment. This statement corresponds to Figure 3B, D, and E. We have described this in the discussion section (page 12, line 392).

  1. I believe that the words “supernatant” in page 10 indicate Af-S. You would unify the style.

Response) We appreciate your suggestion. We revised “supernatant” to “Af-S” per your remark.

  1. Page 10: NAT revealed EV-like particles of around 30–130 nm in the Tim-4 affinity-purified fraction (Figure 3B, D, and E). Please clarify NAT?

Response) We thank you for identifying this error. NAT was corrected to TEM analysis (page 12, line 375).

  1. Page 10: Because gp46 was confirmed in the supernatant fraction obtained through Tim-4 affinity immunoprecipitation, most virus particles were not pulled down by this method and were present in the supernatant fraction. I suggest conducting infectivity assay with the EVs in UFF, Af-S and Af-F to test physiological conditions of the EVs.

Response) We appreciate your insightful suggestion. We would like to conduct the infectivity assay with EVs purified by our method in the future. It would be interesting to clarify whether EVs serve as HTLV-1 infection vectors. Such research could lead to the discovery of a novel transmission mechanism of HTLV-1 via EVs derived from HTLV-1-infected cells. We described this point as limitation of our study (page 14, lines 499–503).

Reviewer 3 Report

Comments and Suggestions for Authors

Kawano et al. in their manuscript titled “Purification method of extracellular vesicles-derived from human T-cell leukemia virus type 1-infected cell without virions” apply a method (Tim-4 immunoprecipitation followed by polymer precipitation) for purifying extracellular vesicles (EVs) from the HTLV-1-infected T-cell line MT-2. Using this new method, the authors find EV fractions did not contain viral glycoprotein Env (as previously published), but did contain gag protein (p24) and HTLV-1 viral genome and tax transcript. Several comments are provided below (major and minor) to help improve the clarity and quality of the manuscript:

Major Points:

1.     Please provide a hypothesis or rationale for why viral p24 capsid protein is found in EVs.  Is this simply random given the abundance of p24 or is there a physiological relevance for the host/virus?  Are other viral capsid proteins (p19 and p15) also present in the EVs (Figure 4)?  Antibodies to p19 exist and could be used to determine this. 

2.     Figure 3: To the non-TEM expert, it is hard to discern what a particle with spike structure looks like. It would be helpful to show EVs contaminated with virus as a control to demonstrate.  It appears that neither Tim-4 affinity IP and UFF followed by TEI have any contaminating virus.

3.     Figure 5 is quite small and difficult to read. Were any HTLV-1 proteins found by mass spec in the EVs obtained through Tim-4 affinity IP?  This should be performed to confirm results in Figure 4. 

4.     In this manuscript, the authors are saying that their method is superior to previously published methods and these previous methods could have virus contamination. To this reviewer, it is difficult to separate the idea that they have further purified EVs into a separate fraction (those which are Tim-4 positive) or have actually eliminated HTLV-1 virions.  

Minor Points:

1.     Abstract, first sentence: Virus-infected cells are not the only cells that produce extracellular vesicles. As written, this sentence could be interpreted as such.  We recommend changing the first sentence to: ‘To mediate intercellular communication, cells produce extracellular vesicles (EVs).’

2.     Introduction

a.     Missing a citation(s) for the following sentence: ‘Four transmembrane proteins called tetraspanins are abundantly….are considered general EV markers.’

b.     EVs have also been found in the CSF, which is of interest to the HTLV-1 field. 

c.     Missing a citation(s) for the following sentence: ‘HTLV-1-infected cell-derived EVs activate T cells.’

3.     Materials and Methods

a.     Figure 1 would fit better in the Results section of the manuscript than in the Materials and Methods. 

b.     Please clarify if the UFF followed by TEI method is exactly the same to past publications in the HTLV-1 field or if it is somewhat different. 

c.     Section 2.3: please correct the phrase ‘followed was as follows’. 

4.     Results:

a.     Figure 4 legend. ‘envelop’ is misspelled and should be ‘envelope’. 

b.     Figure 6: The 7.7kb band in panel C is too faint and difficult to see, even in the positive control land. 

Author Response

Response to Reviewer’s comments.

Dear Editor and Reviewers,

We thank you for your careful review and your pertinent and insightful comments. We have revised our manuscript following the suggestions of reviewer. New references have been added in the revised version, and the reference numbering has been updated accordingly.

The revised sections have been highlighted using yellow lines. We hope that our edits and the responses provided below satisfactorily address all issues and concerns raised by you and the reviewers.

Best regards,

Kunihiko Umekita

  • Reviewer 3

Kawano et al. in their manuscript titled “Purification method of extracellular vesicles-derived from human T-cell leukemia virus type 1-infected cell without virions” apply a method (Tim-4 immunoprecipitation followed by polymer precipitation) for purifying extracellular vesicles (EVs) from the HTLV-1-infected T-cell line MT-2. Using this new method, the authors find EV fractions did not contain viral glycoprotein Env (as previously published), but did contain gag protein (p24) and HTLV-1 viral genome and tax transcript. Several comments are provided below (major and minor) to help improve the clarity and quality of the manuscript:

Major Points:

  1. Please provide a hypothesis or rationale for why viral p24 capsid protein is found in EVs.  Is this simply random given the abundance of p24 or is there a physiological relevance for the host/virus?  Are other viral capsid proteins (p19 and p15) also present in the EVs (Figure 4)?  Antibodies to p19 exist and could be used to determine this. 

Response) We thank you for your insightful suggestion. First, we detected HTLV-1 p19 protein in UFF and Af-F samples. We added information on the anti-HTLV-1 p19 protein antibody in the materials and methods section (page 4, lines 145–146). The revised Figure 4 presents p19 levels in UFF and Af-F samples. The Figure 4 legend was also revised. In the results section, we described the western blotting data for HTLV-1 p19 (page 7, lines 249–252). In addition, we discussed p19 and p24 proteins accumulated by EVs (page 13, lines 418–430). We indicated the possibility that HTLV-1 matrix protein and genomic RNA are internalized into EVs during EV formation in HTLV-1-infected cells (page 14, lines 466–472).

  1. Figure 3: To the non-TEM expert, it is hard to discern what a particle with spike structure looks like. It would be helpful to show EVs contaminated with virus as a control to demonstrate.  It appears that neither Tim-4 affinity IP and UFF followed by TEI have any contaminating virus.

Response) As you indicated, we could not observe any virions or VLPs displaying spike structures in Tim-4 affinity IP and UFF samples under TEM observation. As it was currently difficult to prepare EVs contaminated with viruses as control samples, we could not provide impressive TEM images of EVs and viruses. In addition, we believe there was a limitation regarding negative-stain TEM for observing HTLV-1 virus particles. Prior research have described the difficulty in observing the spike structure of HTLV-1 virions using this technique (Arch Virol. 1982;73:69-73). Therefore, in this study, we performed both TEM analysis and immunoblotting, and confirmed the expression of HTLV-1-related proteins in the Tim-4 affinity IP sample and its Af-S sample. We have described these points in the discussion (page 12, lines 383–390) and limitations (page 14, lines 484–491) sections.

  1. Figure 5 is quite small and difficult to read. Were any HTLV-1 proteins found by mass spec in the EVs obtained through Tim-4 affinity IP?  This should be performed to confirm results in Figure 4. 

Response) We appreciate your suggestion. First, Figure 5 has been enlarged to enhance clarity for readers. Proteome analysis of the EV fraction obtained through Tim-4 affinity immunoprecipitation was performed to evaluate the purity of EVs in Af-F and Af-S fraction (page8, lines 260-261). In addition, most well-known EV marker proteins such as CD81 and CD9 were confirmed in Af-F samples through proteomic analysis. Conversely, Af-S samples appeared to contain ribosomal proteins. Ribosomal proteins are recommended as a purity control for EVs in the MISEV2018 guidelines [11] (page8, lines 264-265). Therefore, proteomic analysis confirmed that the combination of Tim-4 affinity method and polymer precipitation method can collect and purify EVs from HTLV-1 cell culture medium (page8, lines 269-271). Unfortunately, because Proteome Discoverer Software did not support the detection of HTLV-1-related proteins such as gp46, p19, and p24, we could not confirm these HTLV-1-related proteins by proteomic analysis.

  1. In this manuscript, the authors are saying that their method is superior to previously published methods and these previous methods could have virus contamination. To this reviewer, it is difficult to separate the idea that they have further purified EVs into a separate fraction (those which are Tim-4 positive) or have actually eliminated HTLV-1 virions.

Response) Thank you for your comment. To address your comment, the text was revised as follows: “Several studies reported that EVs derived from HTLV-1-infected cells contain HTLV-1-associated proteins and their mRNAs [6,16,17]. Ultracentrifugation and the Nanotrap® (NT) particle method were previously used for EV isolation, but HTLV-1 virions may be present in the purified EV fraction obtained using these methods [16]” (page 2, lines 65–69). The study results suggested that our proposed protocol for purifying EVs can avoid contamination by the virus because Tim-4 selectively binds to phosphatidylserine exposed on EVs. It was unclear whether phosphatidylserine is present in HTLV-1 virus particles. However, our study demonstrated that almost no gp46 envelope protein was detected in Af-F samples collected using the Tim-4 affinity method. Therefore, it can be inferred that almost no phosphatidylserine exists in the HTLV-1 envelope. We have described these points in the discussion section (page 13, lines 431–441).

We believe that EVs collected using the Tim-4 affinity IP method in this study can avoid or greatly reduce virus particle contamination.

Minor Points:

  1. Abstract, first sentence: Virus-infected cells are not the only cells that produce extracellular vesicles. As written, this sentence could be interpreted as such.  We recommend changing the first sentence to: ‘To mediate intercellular communication, cells produce extracellular vesicles (EVs).’

Response) We have revised this sentence according to your recommendation.

  1. Introduction
  2. Missing a citation(s) for the following sentence: ‘Four transmembrane proteins called tetraspanins are abundantly….are considered general EV markers.’

Response) We have cited a reference for this sentence (page 1, lines 32–34).

  1. EVs have also been found in the CSF, which is of interest to the HTLV-1 field.

Response) Per your remarks, we stated that EVs have also been found in the CSF of patients with HTLV-1-associated myelopathy (page 1, lines 39–40).

  1. Missing a citation(s) for the following sentence: ‘HTLV-1-infected cell-derived EVs activate T cells.’

Response) We have cited a reference for this sentence (page 2, line 64).

  1. Materials and Methods
  2. Figure 1 would fit better in the Results section of the manuscript than in the Materials and Methods. 

Response) We thank you for your suggestion. By presenting this flow diagram (Figure 1) in the Materials and Methods, we thought it would be possible to present an overview of the entire experimental protocol and methods used in this research. We hope this layout will help readers understand our research.

  1. Please clarify if the UFF followed by TEI method is exactly the same to past publications in the HTLV-1 field or if it is somewhat different. 

Response) We appreciate your comments. To the best of our knowledge, no previous study used ultrafiltration followed by total exosome isolation for EV purification in the HTLV-1 field.

  1. Section 2.3: please correct the phrase ‘followed was as follows’. 

Response) We have revised this phrase (page 4, lines 128)

  1. Results:
  2. Figure 4 legend. ‘envelop’ is misspelled and should be ‘envelope’. 

Response) We have revised the legend of Figure 4.

  1. Figure 6: The 7.7kb band in panel C is too faint and difficult to see, even in the positive control land. 

Response) We thank you for these comments. We have replaced the image with a higher resolution image (Figure 6D). Meanwhile, the lettering of Figure 6 has been changed because of the addition of a new panel.

Reviewer 4 Report

Comments and Suggestions for Authors

The study by Kawano and colleagues focused on the description of a new method to separate viral particles and EVs.  The authors have used proteomics, TEM and WB analyses to provide evidence to support their claim. However, this reviewer has major concerns, and these are listed below.

1-No comparisons are made to other protocols that have been previously published to separate virus particles from EVs.  One clear example is the Optiprep velocity gradient and the authors should provide data on how both compare.

2-It is not clear why Tim-4 affinity purification was used. Although some EVs contain phosphatidylserine on their outward surface, not all EVs show this characteristic. More troublesome is the fact that no information is available as to the phosphatidylserine content of HTLV-1. The authors do indicate that HIV-1 present this lipid at its surface.  This is thus counterintuitive, and the authors should minimally analyze the lipidic composition of HTLV-1 particles.

3- EM data are not convincing. It is not clear what is intended and the proposed identification of vesicles with spikes (i.e. viral particles) is not efficient. TEM could be complemented with immunoelectron microscopy analyses.

4-Figure 4: WB analyses depict the different viral proteins present in the different types of isolated EV. The authors show that gp46 is solely present in the non-immunoaffinity-purified EVs. Given that this protein is often shedded from viral particles, it would be important to address this possibility of the presence of gp46-negative viruses in the Af-F fraction.  The TM subunit should be similarly analyzed by WB. Alternatively, a negative control could consist of non-specific IgG in the immune affinity step.

5-It is not clear what the results in Figure 5 adds to the study, especially in terms of the ribosomal proteins.  These data should better be explained. What would be the expected results from these proteomics analyses for viral particles?

Figure 6: Detection of RNA in EVs should be quantified and analyzed by RT-PCR.

Comments on the Quality of English Language

Only a few minor mistakes are present in the text. Overall, the quality is acceptable.

Author Response

Response to Reviewer’s comments.

Dear Editor and Reviewers,

We thank you for your careful review and your pertinent and insightful comments. We have revised our manuscript following the suggestions of reviewer. New references have been added in the revised version, and the reference numbering has been updated accordingly.

The revised sections have been highlighted using yellow lines. We hope that our edits and the responses provided below satisfactorily address all issues and concerns raised by you and the reviewers.

Best regards,

Kunihiko Umekita

  • Reviewer4

The study by Kawano and colleagues focused on the description of a new method to separate viral particles and EVs.  The authors have used proteomics, TEM and WB analyses to provide evidence to support their claim. However, this reviewer has major concerns, and these are listed below.

1-No comparisons are made to other protocols that have been previously published to separate virus particles from EVs.  One clear example is the Optiprep velocity gradient and the authors should provide data on how both compare.

Response) In this study, we compared the polymer precipitation method using total exosome isolation (TEI) reagent and the Tim-4 affinity method. Polymer precipitation method is an efficient and simple method to collect EVs such as exosomes. As a result of Western blot (WB) in this study, gp46-positive vesicles were present in both the UFF and Af-S fraction collected by the polymer precipitation method, and it was concluded that both EVs and HTLV-1 particles may exist in these fractions (Figure 4). On the other hand, as a result of WB, gp46 was not detected in the Af-F fraction collected using the Tim-4 affinity method (Figure 4). Therefore, we concluded EVs derived from HTLV-1-infected cells were pulled-down using Tim-4 affinity method without contamination of HTLV-1 particles. Regarding density gradient centrifugation, which you proposed, it has been already pointed out that there is a possibility that virus particles may be mixed in the EVs fraction collected by this method [8,25,26] (page 12, lines 352–354. page 1, line 42). Furthermore, a study in which EVs derived from HTLV-1-infected cells were collected using density gradient centrifugation suggested that a portion of the collected fraction may contain virus particles [27] (page 12, lines 354–357).

2-It is not clear why Tim-4 affinity purification was used. Although some EVs contain phosphatidylserine on their outward surface, not all EVs show this characteristic. More troublesome is the fact that no information is available as to the phosphatidylserine content of HTLV-1. The authors do indicate that HIV-1 present this lipid at its surface.  This is thus counterintuitive, and the authors should minimally analyze the lipidic composition of HTLV-1 particles.

Response) As you pointed out, we have listed as a limitation that the EVs collected using the Tim-4 affinity method may be part of the EVs produced from HTLV-1-infected cells (page 14, lines 495–497). We also described why we verified the Tim-4 affinity method rather than the immunoprecipitation method targeting typical exosome proteins such as CD9, CD81, and CD63 as follow: Immunoprecipitation targeting CD9, CD63, and CD81 exposed on the EV membrane is a specific collection method targeting the membrane. Collection methods targeting these EV marker proteins cause EV damage during EV elution. Therefore, to analyze the biological functions of EVs, a collection method that causes less damage to EVs must be used [8,26]. On the other hand, Tim-4 affinity immunoprecipitation can collect intact EVs [28] (page 12, lines 362– 367).

In this study, we cannot present data on the lipid components of HTLV-1particles. Furthermore, as far as we know, there seems to be no report on the lipid components of HTLV-1 particles. Instead, we focused on the differences in the transmission mechanisms of HTLV-1 and HIV-1 virus particles, and considered the possibility that there are differences in the lipid components of virus particles (page 13, lines 431–441).

3- EM data are not convincing. It is not clear what is intended and the proposed identification of vesicles with spikes (i.e. viral particles) is not efficient. TEM could be complemented with immunoelectron microscopy analyses.

Response) We have described the reason why we attempted to observe EVs and virus particles using negative staining TEM in this study (page 12, lines 383–390). As you pointed out, negative staining TEM may not be a very efficient method for observing EVs and virus particles. However, we believe that it is important to present morphological images of EVs collected using the Tim-4 affinity method in this study. Regarding the observation of EVs using immunoelectron microscopy analyses, which you suggested, please forgive us for the limitations of our ability to do so as we do not have the technology to do so. In addition, regarding immunoelectron microscopy analyzes targeting HTLV-1 viral proteins such as gp46, p24, and p19, we considered the following. As a result of WB, gp46 was not detected in Af-F, but was detected in Af-S and UFF. Therefore, it may be difficult to observe gp46-positive vesicles in the Af-F fraction in immunoelectron microscopy analyzes using gp46 antibodies. Additionally, p19 and p24 were detected in the Af-F fraction. Previous studies have also suggested that EVs contain p19 and p24 (page 13, lines 418–430). Therefore, we considered the possibility that virus particles and EVs could not be distinguished by immunoelectron microscopy analyses using antibodies against p19 or p24 in the Af-F fraction. We have described the limitations of morphological observation and identification of EVs and virus particles by TEM (page 14, lines 484–491). The points you have pointed out are very important. In the future, it will be necessary to conduct more detailed analysis using immunoelectron microscopy of EVs derived from HTLV-1-infected cells collected using our proposed method.

4-Figure 4: WB analyses depict the different viral proteins present in the different types of isolated EV. The authors show that gp46 is solely present in the non-immunoaffinity-purified EVs. Given that this protein is often shedded from viral particles, it would be important to address this possibility of the presence of gp46-negative viruses in the Af-F fraction.  The TM subunit should be similarly analyzed by WB. Alternatively, a negative control could consist of non-specific IgG in the immune affinity step.

Response) As you might expect, gp46 in some virus particles may be shed and lacked. Although given that gp46 is often shed from viral particles, it would be important to address this possibility of the presence of gp46-negative viruses in the Af-F fraction, it is unlikely that all gp46 was lacking from the virus particle. If HTLV-1 particles partially lacking gp46 were collected using the Tim-4 affinity method, gp46 should be detected in the Af-F fraction. However, gp46 was not detected in the Af-F fraction, and gp46 was detected in the Af-S fraction at the same level as UFF. Therefore, in this study, it was thought that EVs other than virus particles were collected in the Af-F fraction, but it could not be denied that virus particles that completely lack gp46 exist in the Af-F fraction. We described these points in discussion (page 12, line 398 – page 13, line 406). In addition, we described the possibility that virus particles lacking gp46 exist in the Af-F fraction as a limitation (page 14, lines 491–493).

About detection of TM subunit, the antibody for TM subunit (gp21) is not commercially available and cannot be purchased, so additional experiments could not be conducted.

5-It is not clear what the results in Figure 5 adds to the study, especially in terms of the ribosomal proteins.  These data should better be explained. What would be the expected results from these proteomics analyses for viral particles?

Response) The purpose of proteomic analysis in this study was to evaluate the purity of EVs in the Af-F and Af-S fractions obtained by Tim-4 affinity immunoprecipitation (Page 8, lines 260-261). Ribosomal proteins are recommended as a purity control for EVs in the MISEV2018 guidelines [11] (page 8, lines 264–265). Proteome analysis showed that the Af-S fraction, which contains many ribosomal proteins, has low EV purity. On the other hand, in the Af-F fraction, the amount of EV marker proteins such as CD81 and CD9 was high, indicating that EVs could be purified to a high degree of purity using Tim-4 affinity method (page 13, lines 442–445). Therefore, proteomic analysis confirmed that the combination of Tim-4 affinity method and polymer precipitation method can collect and purify EVs from HTLV-1 cell culture medium (page 8, lines 269–271).

Figure 6: Detection of RNA in EVs should be quantified and analyzed by RT-PCR.

Response) Regarding Figure 6, the purpose of RNA analysis was to examine whether EVs contained defective and complete HTLV-1 genomic RNA. Qualitative long PCR was performed to easily evaluate defective and complete HTLV-1 genomic RNA based on band size. In addition, we present sequence analysis of HTLV-1 genomic RNA contained in EVs in the supplementary materials. Regarding reverse transcriptase (RT)-quantitative PCR of Tax/Rex and HBZ mRNA, it was unclear which mRNA contained in EVs was appropriate as an internal control mRNA for RT-quantitative PCR. Therefore, we only performed qualitative PCR for Tax/Rex and HBZ mRNA. We thought that this study showed that at least EVs derived from HTLV-1-infected cells contain Tax mRNA.

Round 2

Reviewer 3 Report

Comments and Suggestions for Authors

All requested revisions have been made.  Thank you.

Reviewer 4 Report

Comments and Suggestions for Authors

My concerns have been answered appropriately.